# A Universal Model for the Log-Normal Distribution of Elasticity in Polymeric Gels and Its Relevance to Mechanical Signature of Biological Tissues

**DOI:** 10.3390/biology10010064

**Published:** 2021-01-18

**Authors:** Arnaud Millet

**Affiliations:** 1Team Mechancobiology, Immunity and Cancer, CNRS UMR5309 Inserm U1209, Institute for Advanced Biosciences, 38700 La Tronche, France; arnaud.millet@inserm.fr; 2Grenoble-Alpes University, 38700 La Tronche, France; 3Clinical Research Department, University Hospital of Grenoble-Alpes, 38700 La Tronche, France

**Keywords:** log-normal distribution, atomic force microscopy, universal law

## Abstract

**Simple Summary:**

Mechanical properties of biological tissues are increasingly recognized as important in biology. Atomic force microscopy (AFM) is one of the main tools used to assess elastic properties of various types of biological samples. It has been noted that elasticity values frequently follow a log-normal distribution. We propose in this communication a physical model explaining this fact, and we propose that distribution-type analysis could increase the information obtained from AFM studies on biological tissues.

**Abstract:**

The mechanosensitivity of cells has recently been identified as a process that could greatly influence a cell’s fate. To understand the interaction between cells and their surrounding extracellular matrix, the characterization of the mechanical properties of natural polymeric gels is needed. Atomic force microscopy (AFM) is one of the leading tools used to characterize mechanically biological tissues. It appears that the elasticity (elastic modulus) values obtained by AFM presents a log-normal distribution. Despite its ubiquity, the log-normal distribution concerning the elastic modulus of biological tissues does not have a clear explanation. In this paper, we propose a physical mechanism based on the weak universality of critical exponents in the percolation process leading to gelation. Following this, we discuss the relevance of this model for mechanical signatures of biological tissues.

## 1. Introduction

Following the tremendous progress in research on the pathological processes involved in cancer [1], the mechanical properties of the tumor microenvironment have been increasingly recognized as key parameters in cancer biology [2]. During the progression from a healthy state to malignancy, the mechanical properties of tissues change significantly, and a tumor is seen to interact mechanically with its surroundings during growth. It has been shown that the mechanical forces acting on cells can regulate the signaling pathways responsible for cell death, division, differentiation and migration [3]. At the molecular level, we know that the extracellular matrix is structurally modified during tumor initiation and progression, leading to mechanically modified responses [4]. Recent research has also shown that mechanical phenotyping is an effective quantitative biomarker in cancer biology. Several methods have been used to study the mechanical properties of cells or tissues such as micropipette aspiration [5], laser tweezers [6], magnetic probes [7] and indentation-type atomic force microscopy (IT-AFM) [8]. Due to its superior precision and simple sample preparation, AFM is considered to be more reliable and accurate in analyzing the mechanical properties of tissues at the cell level. Ultrasonic elastography also provides a clear alternative at a larger scale; however, we will focus on cellular mechanobiology in this research. AFM experiments provide spatial maps of elasticity, and the elastic modulus distribution is usually presented in histograms. The question of the extraction of relevant biological information from this data is important. Indeed, it has been proposed that AFM could be used to obtain a nanomechanical signature of cancerous tissues [8]. Further, we have noticed that AFM data of biogels (in this article) or cells [9] frequently present a log-normal distribution. This pattern is a general feature of biological variables as log-normal distributions have been noticed in many biological phenomena [10]. The variety of phenomena calls for a specific model in each case. In this paper, we propose a physical model that could explain the observed log-normal distribution of elastic moduli in biopolymeric gels constitutive of the extracellular matrix.

## 2. Materials and Methods

### 2.1. Collagen Gel

3D scaffolds were generated with Collagen I from rat tail (Life Technologies, Carlsbad, CA, USA). Gelation was performed according to the manufacturer recommendations as previously described [11]. Collagen was diluted at 2 mg/mL in PBS (Phosphate Buffer Saline, Life Technologies, Carlsbad, CA, USA) and pH at 7.0 was obtained using sterile NaOH (Merck Millipore, Burlington, MA, USA). Collagen is then incubated at 37 °C in a humidified incubator for 40 min. Gels were gently rinsed with PBS for 15 min before cell seeding.

### 2.2. Cell Culture

Human blood samples from healthy de-identified donors were obtained from EFS (French National Blood Service, Grenoble, France) as part of an authorized protocol (CODECOH DC-2018–3114). Donors provided signed consent for the use of their blood in this exploratory study. Peripheral blood mononuclear cells (PBMCs) were obtained from whole blood (leukocyte reduction system cones) by density gradient centrifugation (Histopaque 1077, Merck Millipore, Burlington, MA, USA). Monocytes were then isolated from PBMCs using CD14 magnetic beads (Miltenyi Biotec, Paris, France) according to the manufacturer’s instructions, as previously described [11]. Purity was assessed by flow cytometry for CD14hiCD45hi cells and found to be more than 96%. The monocytes were cultured in RPMI-Glutamax (Life Technologies, Carlsbad, CA, USA) supplemented with 10% human serum AB (Merck Millipore, Burlington, MA, USA) and differentiation was induced by M-CSF (25 ng/mL) over 6 days. 3D conditions were obtained by seeding the monocytes on the top of gels, and cells subsequently invaded the gel during differentiation. Polarization was later obtained using the same concentration of M-CSF and adding specific stimulations for a total of 48 h. Polarization states were induced as follows: M1 macrophages IFNγ 10 ng/mL + LPS 1 ng/ml and M2 macrophages IL-4 20 ng/mL + IL-13 20 ng/mL. All cytokines and growth factors are purchased from Miltenyi Biotec and LPS serogroup 0111 B4 was purchased from Calbiochem (Merck Millipore, Burlington, MA, USA).

### 2.3. Atomic Force Microscopy

We used the FlexFM Atomic Force Microscope (Nanosurf, Liestal Switzerland; ARTIDIS version). Borosilicate beads (radius 5 μm) mounted on QP-SCONT cantilevers (NanoAndMore, Wetzlar, Germany) were also used. The nominal spring constant was 0.01 N/m and was specifically assessed using the Sader method [12]. The force setpoint was fixed at 2 nN. Elasticity was determined from the retraction curve using the AtomicJ software using a force-volume mode (indentation mode) [13]. The measures were done in acellular areas identified thanks to a camera in order to measure the ECM elasticity alone. AFM measurements were done in PBS (Phosphate Buffer Saline). Since adhesion was negligible compared to our force set point and indentation was far less than our tip radius, we used a Hertz-model to relate the force and the indentation. In this model, the force *F* is related to the indentation *δ* and the Young’s modulus *E* by the following relation
(1)F=4RE3(1−υ2)δ32
where *R* is the radius of the tip and *υ* is the Poisson ratio taken at 0.5.

## 3. Results

### 3.1. Elastic Moduli of Collagen Based Scaffolds Displayed a Log-Normal Distribution

The elastic moduli of cells have been shown to present a log-normal distribution [9]. We hypothesized that this general feature could be extended to the extracellular matrix. Collagen type I is the main component of the extracellular matrix in mammals. We used the possibility to control the sol-gel transition by pH and temperature to obtain a collagen gel. This biomimetic hydrogel was secondarily analyzed using IT-AFM. The distribution of the elastic modulus obtained by AFM is represented in Figure 1 in linear and log scales.

The log-normal distribution of elasticity was confirmed by the Kolmogorov–Smirnov test (linear distribution) and a Shapiro–Wilk normality test (log-scale distribution), illustrating the general feature of log-normal distribution in biological tissues. Considering the simplicity of the physical properties of a collagen gel compared to real biological tissues, we propose a physical model of the observed distribution. Our model, based on gel physics, differs from previous ones. To the best of our knowledge, previous general explanations of the log-normal distribution in biology have not been applied to biopolymers. Moreover, we find that they do not provide a clear understanding of the physics involved.

### 3.2. Log-Normal and Normal Distribution Laws

To better understand why log-normal distributions could appear in natural variables, we must first understand the circumstances under which the “expected” normal distribution is relevant. The central limit theorem of probability can be formulated in the following general form [14]:

Given ${\left(X_{n}\right)}_{n\geq 1}$ a collection of independent random variables, we note $\sigma_{n}{}^{2}$ as their variance and suppose that $\sum \sigma_{n}$ diverges. Then, there exist two positive real numbers ε and M such that for each random variable the expectation value follows the inequality $E\left({\left|X_{n}\right|}^{2+\varepsilon }\right)\leq M$. Then
(2)$$\frac{\sum_{j=1}^{j=n}\left(X_{j}-E\left(X_{j}\right)\right)}{\sqrt{\sum_{j=1}^{j=n}\sigma_{j}{}^{2}}}\overset{convergence\;in\;law}{\to }N\left(0;1\right)$$
where N(0;1) is the normal law of mean 0 and variance 1. The convergence in law refers to the weak convergence of probability distribution which only requires the convergence of the expected values of the random variable collection. This theorem relies on the hypothesis of the independence of random variables. Moreover, this general presentation has the advantage of removing the more classical condition that each random variable should have the same distribution law. The way in which normal laws appear experimentally from this result could be explained in the following manner:

By studying a random variable X, the successive values measured could be described by a recursive process of the type as follows
(3)$$X_{j+1}=X_{j}+R_{j}$$
where R is the random error for each step that is supposed to be normally distributed and independent of X. This could lead to a normal law distribution as we can write
(4)$$X_{n}=\;X_{0}+\overset{j=n}{\underset{j=1}{\sum }}R_{j}$$

Using the fact that a sum of independent normal variable is a normal variable itself. In the case where the independence of variables no longer applies, the simple model of recursive process could take the following form where each step is dependent on the previous one [15]:(5)$$X_{j+1}=X_{j}\left(1+R_{j}\right)$$
which gives
(6)$$X_{n}=X_{0}\overset{j=n}{\underset{j=1}{\prod }}\left(1+R_{j}\right)$$

Considering the logarithm of this relation and the fact that the error is always small in magnitude, we find
(7)$$\ln\left(X_{n}\right)\;\approx \ln\left(X_{0}\right)+\;\overset{j=n}{\underset{j=1}{\sum }}R_{j}$$
which gives a normal distribution for ln(X) and following this, a log-normal distribution for the experimental variable.

### 3.3. Log-Normal Distribution in Biology: A Heuristic Argument

In order to justify the dependence between random variables used in Equation (5), various heuristic arguments have been proposed. These arguments rely on kinetic models of biochemical reactions [16], which in turn can be described generically by the following form:(8)$$\frac{dc\left(t\right)}{dt}=\left(\bar{a}+\;\delta \left(t\right)\right)\;c\left(t\right)$$
where c(t) represents the concentration of a biomolecule, which will be our experimental variable. In order to described the noisy kinetics found in the real world, a fluctuation term δ(t) which follows a normal law, has been introduced. This equation will give a (continuous) recursive relation similar to Equation (7) in the following form
(9)$$\frac{dln\;\left(c\left(t\right)\right)}{dt}=\bar{a}+\;\delta \left(t\right)$$

Equation (9) will thus give a log-normal distribution for the concentration c(t).

While these arguments can justify the presence of the log-normal distribution in cellular biology, they do not rely on clear physical processes and are unable to provide a convincing explanation for log-normal distribution of elasticity in biologically relevant gels.

### 3.4. The Heuristic Argument is Unable to Explain the Log-Normal Distribution for Gels

We can apply Equation (9) to describe the polymerization of fibers. In order to do so, the concentration of nodes, c(x,t), will also depend on the various positions of the nodes of the network. Now $\bar{a}$ depends on the position of the nodes. Spatial fluctuations from Brownian noise will give a normal distribution for $\bar{a}$. We will then obtain a density of nodes that will present a spatial normal distribution. The elastic modulus of a polymeric gel could be related to the density of nodes of the network [17]
(10)E≅ckTQ−13
where k is the Boltzman constant, T the temperature and Q the swelling ratio. The resulting elastic modulus will not present a log-normal distribution according to our heuristic argument, contrary to what is found experimentally (Figure 1). We conclude that a physical explanation is needed and that this heuristic argument is unable to provide a model for the log-normal distribution of elastic modulus values found in polymeric gels.

### 3.5. Percolation Model of the Elastic Modulus of A Polymeric Gel

Percolation networks can provide comprehensive physical models of gels mechanical properties. Similar to percolation models of electric conductive networks, the elastic moduli of polymeric gels could be described using a percolation model of the sol-gel transition. This model predicts the existence of a critical exponent f associated with the elastic modulus [18]. If p is the percolation parameter which is related to the fraction of polymer molecules linked together forming the nodes of the network and $p_{c}$ is the critical percolation threshold, then the elastic modulus E could be expressed as follows:(11)$$E\sim {\left(p-p_{c}\right)}^{f}$$

This model has been validated experimentally [19]. In order to understand the relation between various critical exponents involved in scaling laws, explicit elastic percolation network models have been proposed. For example, Kantor and Webman used a percolation network whose bonds could be stretched or bent [20]. The Hamiltonian of the network could be expressed as follows:(12)$$H=\frac{\alpha }{2}\underset{\langle ij\rangle }{\sum }e_{ij}{\left[\left({\vec{u}}_{i}-{\vec{u}}_{j}\right)\cdot \vec{r_{ij}}\right]}^{2}+\frac{\beta }{2}\underset{\langle ijk\rangle }{\sum }e_{ij}e_{ik}{\left(\delta \theta_{ijk}\right)}^{2}$$
where $e_{ij}$ are random variables taking values 0 or 1 with probability p and 1−p, ${\vec{u}}_{i}$ is the infinitesimal displacement of site i, $\vec{r_{ij}}$ is the unit vector between site i and j, and $\delta \theta_{ijk}$ is the change of angle between the j−i and i−k due to infinitesimal displacement. The first sum is done on every bond and the second one on every triplet of nodes forming bonds with i at the center. This model in 3D could help clarify the importance of the stretching α and the bond-binding β on the values of $p_{c}$ and f. Indeed, numerical simulations have suggested that the critical exponent f is sensitive to molecular details, leading to a weak class of universality [21]. We propose the use of this percolation network model to understand the nature of the distribution of elastic moduli in gels. To apply this model, we need to account for the swelling of the biopolymeric gel. In tissues, interstitial fluid is always present leading to a swelling of the extracellular matrix. When gels swell, the links between nodes of the network are heterogeneously modified, leading to bending and stretching as illustrated in Figure 2.

In the percolation model, this will lead to a spatial dependence of α and β. Further, $p_{c}$ will be less impacted by this process as it is mainly related to the density of bonds during the sol-gel transition before swelling; this parameter does not change after swelling. The modification of bonds due to the swelling will mainly display a spatial dependence for f, which will now present a normal distribution. The characteristic length of these spatial inhomogeneities will be ≫ξ the percolation correlation length (roughly the mean distance between nodes). After swelling, when elastic modulus measurement is performed, we observe a spatial variation of the elastic modulus. Using a logarithmic form of Equation (10) we obtain
(13)$$\log E\;\sim \;f\;log\left|p-p_{c}\right|$$
where $p-p_{c}$ will not contribute to spatial variations as it reflects the density of the nodes of the network not modified during swelling. We then see that the elastic modulus of the gel presents a log-normal distribution.

### 3.6. AFM Analysis of A Simplified Biological Tissue

In accordance with our model, the log-normal distribution is directly related to the mechanical properties of the network. We can hypothesize that modifications to the structure of the network will result in deviation from the log-normal distribution. We propose that deviation from the log-normal distribution could be used to obtain a mechanical signature that translates structural modifications of the network. This possibility is particularly relevant in biological tissues where cells could degrade or remodel their surrounding matrix. Mechanical modifications induced by pathological processes have been used to obtain a mechanical signature of cancerous tissues using AFM in other research [8]. The mean value of the elasticity of cancerous tissues was compared to healthy ones and the comparison with histology analysis of tissues revealed the suitability of this approach [8]. Our model suggests using the type of distribution obtained in the AFM studies of biological tissues to extend the understanding of the mechanical modifications of tissues driven by cellular processes. As the final goal of obtaining a complete mechanical signature of the tissue is not within our reach, we propose, in this article, illustrating the biological relevance of our model in a simpler system. We used human macrophages embedded in a collagen gel to obtain a simplified tissue. Macrophages were chosen for the study as they are found in every tissue and are involved in the remodeling of extracellular matrix [22]. From this setting, we can study the extent to which these cells can modify their surrounding gel network according to their activation. To reproduce the various states of activation found in vivo, we activated these cells to obtain M1 (pro-inflammatory) macrophages, which are prone to express metalloproteinases (such as MMP2, MMP9 or MMP14) that could digest the collagen. We also obtained M2 (anti-inflammatory) macrophages that express TGM2 (transglutaminase 2) which could covalently bond collagen fibers. We then performed an AFM analysis to obtain the elastic moduli of gels. The mean value for M2 macrophages was 307.7 +/− 112 Pa and 267 +/− 151 Pa for M1 macrophages, showing a slight decrease of the elastic modulus in gels containing M1 macrophages (*p* < 0.001 non parametric Mann–Whitney U test). Moreover, we found that M1 macrophages were associated with the loss of the log-normal distribution of elasticity probably due to their degradation activity on the gel contrary to M2 macrophages, which present a log-normal distribution (Figure 3). This result illustrates the added information provided by the determination of the type of distribution in contrast to the comparison of the means alone.

## 4. Discussion

We have proposed a general model accounting for the log-normal spatial distribution of elasticity found in biological polymeric gels. This model relies on a low universality class of critical exponents of the percolation network and applies to every polymeric gel that presents a spatial heterogeneity of its network following swelling, as in the case of biological tissues. The results illustrate the possibility of obtaining structural information concerning the extracellular matrix from the type of distribution of the elastic modulus obtained by AFM. We have illustrated this point using a simplified model of a biological tissue where immune cells could remodel the extracellular matrix network. The main limitation of our study is related to the structural and cellular complexity found in real biological tissues where elastic measures combine ECM and cell mechanical properties. Various types of cellular processes can induce structural damages associated with a deviation from the log-normal distribution. Despite this, our preliminary results suggest that distribution-type analysis could increase the information obtained from AFM studies on biological tissues. This work opens the possibility to obtain relevant structural information of the extracellular matrix on de-cellularized tissues. This result could help to obtain clinical relevant mechanical signatures of pathological tissues, like in cancer where collagen synthesis is frequently found. As the immune response to tumor growth is recognized as a key event [23,24,25,26], we have used a simplified model involving macrophages, which constitute the largest noncancerous cell population in various solid tumors. These cells are involved in the remodeling of the extracellular matrix network and our AFM analysis was able to reveal the signature related to their polarization. This opens the possibility to use AFM signature to analyze the immune response due to macrophages in the tumor microenvironment through their implication in the extracellular matrix remodeling.

Further studies will be required to extend our results to tissues with higher cell density than the simplified model used in this work, and it will be also needed to validate that our result could be extended to cell mechanical signature when cells participate to the elastic measure obtain by AFM. It will also be necessary to demonstrate the correlation between elastic modulus distribution and the structural properties of biological tissues.

## Figures and Tables

**Figure 1 biology-10-00064-f001:**
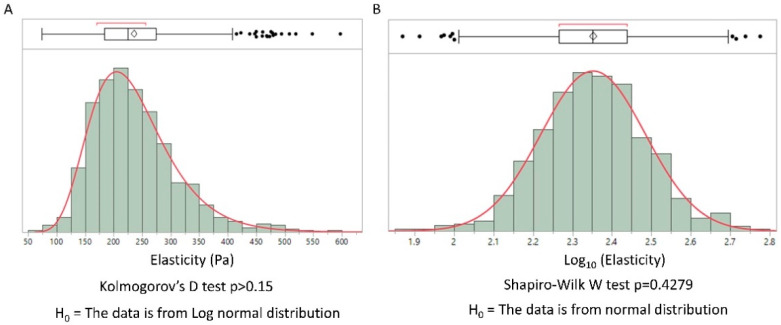
Distribution of elasticity values from atomic force microscopy (AFM) measurement on a collagen type I gel. (**A**) distribution values of elasticity in linear scale, a Kolmogorov’ D test for log normal distribution was performed. (**B**) The same values represented with a logarithmic scale; a Shapiro–Wilk normality test was performed. In each case we found a log-normal distribution. (Histograms were obtained using the JMP software v14.2 SAS).

**Figure 2 biology-10-00064-f002:**
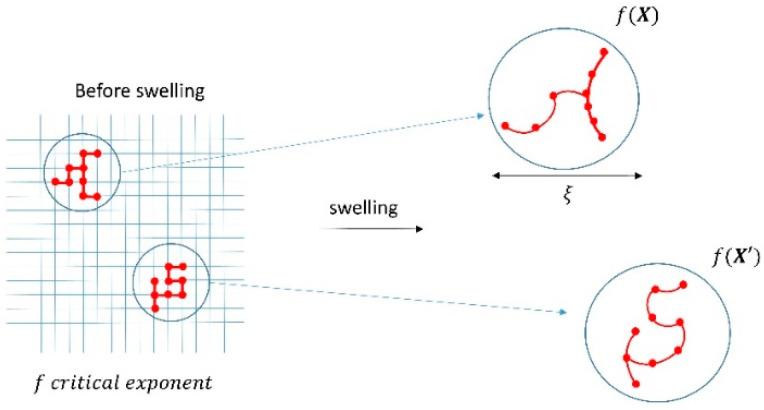
Schematic of the swelling of a gel based on a percolation model. The parameter p related to the number of nodes is not modified by the swelling, but the bending of the links between nodes will impact the critical exponent f that will now depend on the position translating the weak universality.

**Figure 3 biology-10-00064-f003:**
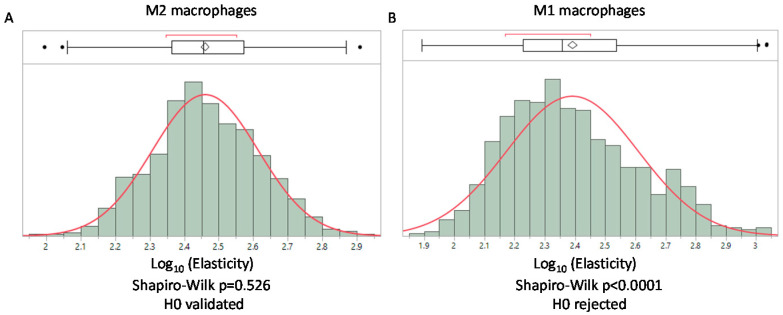
Distribution of elasticity values from AFM measurement on a collagen type I gel with macrophages polarized as M2 and M1. (**A**) log-normal distribution of elasticity for gels containing M2 macrophages. (**B**) Loss of the log-normal distribution when macrophages embedded in the gel are polarized toward M1. In each case a Shapiro–Wilk normality test is performed. (Histograms were obtained using the JMP software v14.2 SAS).

## Data Availability

The data presented in this study are available on request from the corresponding author.

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
