# Peer review of "A Universal Model for the Log-Normal Distribution of Elasticity in Polymeric Gels and Its Relevance to Mechanical Signature of Biological Tissues"

_biology, 2021, doi:10.3390/biology10010064_

Round 1

Reviewer 1 Report

Dear Editor,

I have reviewed manuscript "A universal model for the log normal ... tissues" by Dr. Arnaud Millet. Basically, the author builds on the finding that the universal log-normal distribution of the Young modulus observed via AFM experiments conducted with elastic gels (assimilated to biological tissues).

This behavior cannot be explained by heuristic arguments based of the kinematics of gel growth. Rather, the author adapts a percolation model for elastic gels, cited as reference 18 in the manuscript; (however, the list of references is limited to 11.)  

The novelty of the present study is the use of the universal behavior of the elastic modulus (equation 11, in the text), experimentally observed (reference 19, not listed in the draft). Using a model by Kantor and Webmann (reference 20, not listed), the author claims that the exponent f has a normal spatial distribution, which yields a log-normal distribution for the elastic modulus. Hence the universality of the log-normal distribution for the elastic modulus, since this power dependence (eq 11 or 13) is universal (property of the percolation model).

I have some reservation regarding the hidden assumption beneath both the experimental measurement and the model approach.  

(1) Experimental data. It is assumed that the experimental gel has linear behavior (small deformation, linear constitutive relation, as implied by the use of the Hertz formula).  

(2) Similarly, the model (eq. 12 in the draft) is valid for small node displacement or bond bending. The percolation model does not rely on the bending/stretching of bonds but on the connectivity of bonds.  

Therefore, I suspect that the universal behavior, as addressed in this draft, results from the linear regime of the experimental observation and model analysis.

It would be interesting to check whether the non-linearity of the material (or its anisotropy due to the presence of fibers and elongated structures) change the conclusion drawn from the observations conducted in the linear regime.

This would be of great importance for actual cells, since it is known that actual cells can suffer large deformations (think of a cell going through a compact tissue), implying non-linear behavior of the strain-stress relationship. Additionally, the presence of fibers (stress fibers) and structures in cells would be important to take into account. Therefore, a model study using finite element models, contact mechanics between the AFM tip and the cell, the geometry of the contact, large deformation and hyperelastic model would be a nice addition to the study.

Reviewer 2 Report

This manuscript submitted as a communication to Biology is very well-compiled, however, the scope might be slightly off from that of journal. While the mathematical models are presented to show the log normal behavior of elasticity of extracellular matrix, I have some major concerns on the experimental model which are presented below.

  1. Why did author decide to seed the cells on top of gel and wait for cell to invade the gel matrix during differentiation phase? Encapsulating the cells within the gel can mimic the actual 3D extracellular environment from the onset of cellular processes and makes the author claim of simplified tissue model more relevant.  
  2. I would want the author to clearly specify the sample preparation for AFM studies in the method section. Were the gel samples dried before running AFM studies? Also, please mention the AFM mode applied to fetch the mechanical data?
  3. Please include some insights on how the findings of this study would benefit the cancer research as the introduction section predominates this specific topic. Given the highly dynamic and heterogeneous cancer microenvironment, the experimental model presented here is not competent and relevant.  

Round 2

Reviewer 2 Report

I am satisfied with the author response and would be glad to recommend it for consideration.